# Youths' perceptions and behaviors on COVID-19 testing

**Caleb Arthur**[1]*, **Kathleen Abenes**[1], **Marika Waselewski**[1], **Samantha A. Chuisano**[1], **Tammy Chang**[1,2]

**1** Department of Family Medicine, University of Michigan; Ann Arbor, Michigan, United States of America, **2** Institute for Healthcare Policy and Innovation, University of Michigan; Ann Arbor, Michigan, United States of America

* caarthur@med.umich.edu

## Abstract

### Background

Social gatherings are frequent sources of COVID-19 infections, especially among youth. However, little is known about youth testing behaviors before and after gatherings. Our aim was to assess behaviors and perceptions of youth related to testing for COVID-19 before or after social gatherings in order to inform efforts to reduce disease spread.

### Methods

Five open-ended questions were texted to participants aged 14–24 throughout the United States via MyVoice. Using a content analysis approach, two investigators reviewed responses by question, developed a codebook, and independently applied codes. Discrepancies were resolved via discussion. Code frequency and demographic data were summarized using descriptive statistics.

### Results

Of 1204 participants, 989 responded to at least one question (RR = 94.1%). The mean age was 20.2 years (SD: 2.4 years). Most participants (80.7%) reported testing for COVID-19 at least once. Most (70.6%) were likely to test following an event, especially "[i]f someone at the gathering tested positive," while a smaller number (50.9%) endorsed testing prior to a gathering. Of youth who would not get tested, being vaccinated was the highest reported.

### Conclusion

Youth in our nationwide sample are likely to test for COVID-19 after an event, though less likely if they are vaccinated. Their desire to test is primarily driven by symptoms, exposures, and requirements. Youth are interested in increased access to home testing. Youth-centered communications regarding testing recommendations and increased test availability for youth may reduce COVID-19 spread among young people and inform future pandemic recommendations.

**Data Availability Statement:** The datasets generated and/or analyzed during the current study are not publicly available due the nature of our IRB approval and requirements for Data Sharing Agreements to be executed with any individual or

organization interested in accessing our data. Data is available upon request after completion of a data sharing agreement. Please contact irbmed@umich.edu or 734-763-4768 / (Fax 734-763-1234); 2800 Plymouth Road, Building 520, Room 3214, Ann Arbor, MI 48109-2800 to request access to data.

**Funding:** The author(s) received no specific funding for this work.

**Competing interests:** The authors have declared that no competing interests exist.

## Introduction

The Centers for Disease Control and Prevention (CDC) notes that testing for COVID-19 is critically important to reduce the spread of the virus [1]. Testing for has become widely available several years into the pandemic. In addition to the different types of tests (PCR and rapid antigen tests), testing is also now available in a variety of locations [2]. These sites initially included pharmacies and health centers, but now include home settings [3]. As of January 2022, insurance companies are required to cover the cost of at home PCR and antigen tests to increase access [3]. In 2022, the United States Postal Service (USPS) also allowed up to three free orders of antigen kits per household [4].

In 2022, the CDC updated the recommendation for testing in the following scenarios: if an individual has symptoms of COVID-19 or at least five days after close exposure to an individual with COVID-19, even without symptoms [1]. Later that year, the CDC changed its guidelines, which advised testing based on hospitalization rates of COVID instead of community levels [5]. This recommendation replaced testing prior to attending an indoor event, before and after travel, and for screening purposes for work or school. The CDC's guidelines have varied throughout the pandemic [6] but notably do not account for vaccination status [7, 8]. An analysis of adults in the White House Coronavirus Task Force early in the pandemic revealed 94.0% [9] adherence to avoidance of social events. There have been several studies [10, 11] that examine personalities and demographics surrounding COVID health advisories, but little is known about whether young adults or youth adhere to testing guidelines.

Youth aged 14–24 years represent over 14% [12] of the United States population, and their behaviors have a strong influence in the spread of COVID-19 in their communities [13]. Many youth are often asymptomatic when they contract COVID [14] demonstrated by a 2022 meta-analysis showing that 43.75% of those testing positive under 20 years of age were asymptomatic. Incorporating the experiences and perspectives of youth regarding testing and social gatherings will support the development of effective policies. Therefore, the aim of this study was to assess the behaviors and perceptions of youth related to testing for COVID-19 before or after social gatherings to inform efforts to reduce the spread of COVID-19 as well as future pandemics.

## Methods

Participants in this study were part of MyVoice, a longitudinal nationwide text message poll that seeks to understand youth opinion on salient health and policy issues. MyVoice recruitment occurs online via targeted social media advertisements based on weighted benchmarks from the American Community Survey. Ads are targeted based on age, gender, race and ethnicity, and region of residence to encourage a diverse cohort. Cohorts range typically between 1000 to 1200 participants. Individuals ages 14–24 years with literacy in the English language and access to a phone with texting capabilities are encouraged to apply [15, 16]. Questions posed to MyVoice youth are iteratively developed with a team of youth, survey experts, and adolescent health researchers to ensure clarity and appropriateness of language. This study was approved by the University of Michigan Institutional Review Board (HUM00210274). Written informed consent was obtained electronically from all participants; given minimal risk, parental consent was waived by the IRB.

On November 19, 2021, participants were sent a text message survey with the following five open-ended questions: (1) What, if anything, would prompt you to get tested for COVID-19? (2) Would you ever get tested for COVID-19 BEFORE a social gathering? Why? (3) Would you ever get tested AFTER a social gathering? Why? (4) Have you ever gotten tested for COVID-19? Why? (5) Would access to free COVID-19 self-tests be helpful to you? Why or why not?

Participants had one week to answer the survey before data was downloaded for analysis. Using an inductive approach, two team members individually reviewed fully anonymized youth responses for each question and identified common responses that were then used to develop a codebook [17]. Next, two investigators independently applied codes to individual responses by question for the presence of content (i.e., if a response mentioned multiple COVID symptoms, it would be coded as "symptoms" once). Any disagreements in coding were discussed until consensus was reached. Codes were not mutually exclusive, and many responses were assigned multiple codes. Descriptive statistics were used to illustrate the prevalence of codes in each question [18]. Summary statistics for demographics and code frequencies were completed using Google Sheets.

## Results

Of the 1204 participants, 989 responded to at least one question in the survey (Table 1). Respondents were 20.2 years of age on average (SD: 2.4 years) with a range from 15–24 years. Most respondents identified as male (56.8%) and non-Hispanic white (67.0%) with many having completed some college or technical school at the time of the survey (43.0%) and living in a suburban setting (56.4%). For comparison, 49.5% [19] of the nation identifies as male, 59.3% [20] as non-Hispanic white, and 63.5% [21] as completing some college or technical school. The Census Bureau does not provide data for those in suburban areas, but it is estimated that 80.0% live in urban settings while 20.0% live in rural settings [22].

**Table 1. Respondent demographic characteristics (n = 989).**

| | n (%) or mean (SD) |
|---|---|
| **Age** | 20.2 +/- 2.4 |
| **Gender** | |
| Male | 562 (56.8) |
| Female | 336 (34.0) |
| Other gender identity (i.e., nonbinary, transman, transwoman, etc.) | 91 (9.2) |
| **Race and ethnicity** | |
| Non-Hispanic White | 662 (67.0) |
| Hispanic | 92 (9.3) |
| Non-Hispanic Black | 62 (6.3) |
| Asian | 117 (11.8) |
| Mixed Race/Other | 58 (5.9) |
| **Education level** | |
| Less than high school | 221 (22.4) |
| High school graduate | 134 (13.6) |
| Some college or technical school | 425 (43.0) |
| Associate's or technical degree | 106 (10.7) |
| Bachelor's degree or higher | 103 (10.4) |
| **Region** | |
| Midwest | 284 (28.7) |
| Northeast | 209 (21.2) |
| South | 274 (27.7) |
| West | 221 (22.4) |
| **Rurality** | |
| Rural | 86 (8.6) |
| Suburban | 554 (56.4) |
| Urban | 343 (34.9) |

The main findings from the content analysis include: (1) Most youth in our sample have been tested for COVID-19, primarily in response to experiencing symptoms associated with the virus, (2) More youth would get tested after a gathering than before, and (3) Access to free home testing is perceived as helpful to the participants.

The majority (80.7%) of our sample have tested for COVID-19 at some point during the pandemic. The most common reasons that would prompt testing include symptoms consistent with or suspicious for, COVID-19 (48.2%, 471 of 970) such as "[f]ever [or] no smell," followed by exposure to confirmed cases or "[s]uspicion of contact with someone infected" (35.9%, 348 of 970), and requirements such as ". . .college is forcing me to" or ". . .required by travel" (9.0%, 87 of 970) (Table 2).

A greater number of participants responded that they would test for COVID-19 after a social gathering (70.6%, 657 of 931) than before one (50.9%, 488 of 958). There were more responses from those who would not get tested before a social gathering (30.5%, 292 of 958) than those who would not get tested after a social event (15.7%, 146 of 931) with vaccination status the most commonly cited reason (18.5%, 86 of 465 and 19.6%, 52 of 265, respectively). Additional reasons for not testing before included lack of involvement in social gatherings (5.6%, 26 of 465), "feeling fine" (9.9%, 46 of 465), and believing that the test is unnecessary (6.7%, 31 of 465). Other reasons for not testing after a social event included mask use in public spaces (14.2%, 11 of 265) and beliefs that testing was pointless or that they would know without testing when they were sick (7.2%, 19 of 265).

Two thirds of respondents (67.5%, 629 of 932) thought access to free at-home tests would be helpful "because it is easier" (12.7%, 118 of 932), allow for "use [of tests] proactively" (9.3%, 87 of 932), and to provide peace of mind (9.4%, 88 of 932). Reasons cited against the utility of at-home tests (21.4%, 199 of 932) included confidence that respondents would not use them (13.7%, 35 of 199), belief these tests are not as accurate (10.6%, 27 of 199), and preference "to be tested by a professional" or that "it is already very easy to get tested" (17.6%, 45 of 199).

## Discussion

Nearly all youth in our sample report having been tested for COVID-19 at some point during the pandemic, largely due to experiencing symptoms. The majority would get tested after a gathering, and fewer reported that they would test before a gathering, especially if the respondents have been vaccinated. Overall, youth in our sample endorse a reactive rather than proactive approach to testing with most youth citing symptoms, exposures, requirements by school or work, and availability as reasons for testing.

Youth in our study have largely been tested but most commonly cite their own vaccination status as a reason to not get tested both before and after a social gathering. While this does suggest trust in COVID-19 vaccines in this age group, current CDC testing guidelines [1] do not vary based on vaccine status and in fact the American Academy of Pediatrics states [7] that vaccination status should not guide testing. While transmission and development of COVID-19 symptoms are reduced with even a single dose [23, 24] of any of the vaccines available in the US, their efficacy does not yet support the assumption that immunization eliminates the need for testing in the instances outlined by the CDC. Education on appropriate timing of testing at the moment youth are getting vaccinated may represent a teachable moment for parents and youth alike [25–27]. Finally, communication that is targeted towards youth themselves may promote more testing as greater access to self-test kits are available.

Most respondents tested for COVID-19 because they experience symptoms or were exposed, but reasons such as school or work requirements were also commonly reported. COVID symptoms are broad, however, so many individuals with COVID-19 are classified as

**Table 2. Questions, codes, and example participant quotes.**

| Question, Code | All n (%)* | Example Participant Quote |
|---|---|---|
| **1. What, if anything, would prompt you to get tested for COVID-19? (n = 970)** | | |
| Symptoms | 471 (48.2) | "If I don't taste anything and I feel sick" |
| Exposure | 348 (35.9) | "If I knew that someone that was around me in the past couple weeks had COVID" |
| Socializing | 103 (10.6) | "Seeing the new my hero academia movie with my best friend" |
| Nothing | 83 (8.6) | "There are no reasons that impel me to do that" |
| Required | 87 (9.0) | "I have to do it because I work in a restaurant." |
| General | 78 (8.0) | "The curiosity to know if I am healthy" |
| Travel | 59 (6.1) | "Avoid being prevented from entering another country for not having a negative Covid-19 test." |
| **2. Would you ever get tested for COVID-19 BEFORE a social gathering? Why? (n = 958)** | | |
| Yes | 488 (50.9) | |
| Depends | 173 (18.1) | "Depends on the gathering and location. " |
| No | 292 (30.5) | |
| *Reasons for testing*** | | |
| Keep others safe | 195 (29.5) | "Yes to make sure I don't transmit it to a large group" |
| Required | 153 (23.1) | "Yes if I need to show a negative result before attending." |
| General | 121 (12.6) | "Absolutely. You can't be too cautious." |
| Symptoms | 81 (18.3) | "yes, if I feel sick even if i am not around people i will get teste" |
| High risk | | 59 (8.9)"Yes, sometime, when I have to meet with my grandparents to make sure they are not in danger" |
| *Reasons against testing*** | | |
| Vaccinated | 86 (18.5) | "I do not believe it. Because I am vaccinated, I should not be a problem for the other people who attend the social gathering." |
| Not sick | 46 (9.9) | "No, because I feel fine and don't have a fever" |
| Unnecessary | 31 (6.7) | "No, because there is no point in my opinion" |
| Access | 27 (5.8) | "No, it takes too much time |
| **3. Would you ever get tested AFTER a social gathering? Why? (n = 931)** | | |
| Yes | 657 (70.6) | |
| Depends | 119 (12.8) | "Maybe" |
| No | 146 (15.7) | |
| *Reasons for testing*** | | |
| Exposure | 267 (34.4) | "Yes if someone who was there has tested positive" |
| General | 218 (28.1) | "I would just to be safe" |
| Symptoms | 171 (22.0) | "Yes of course, if I have symptoms of covid" |
| Location | 83 (10.7) | "Yes, I would do it if the social gathering was in a space with many risks of transmission of covid-19." |
| *Reasons against testing*** | | |
| Vaccinated | 52 (19.6) | "No, because I am vaccinated." |
| Unnecessary | 19 (7.2) | "No, I have taken precautions during the gathering and therefore would not need it." |
| **4. Have you ever gotten tested for COVID-19? Why? (n = 944)** | | |
| Yes | 762 (80.7) | |
| Once | 45 (5.9) | |
| 2+ | 224 (29.4) | |
| No | 182 (19.3) | |
| *Reasons for testing* | | |
| Sick/Symptoms | 210 (27.6) | "Yup. Thought I had symptoms" |
| Contact/Exposure | 185 (24.3) | "Yes because I was in close contact with someone who tested positive at work" |
| Work/School | 164 (21.5) | "I have gotten tested for covid due to my school's policy" |
| Required | 120 (15.7) | "Yes, it was required before my surgery" |
| Routine/Safety | 93 (12.2) | "Yes, just to see if we're ok" |
| *Reasons not tested* | | |

*(Continued)*

**Table 2.** (Continued)

| Question, Code | All n (%)* | Example Participant Quote |
|---|---|---|
| No exposure | 33 (18.1) | "Not yet. Cuz no one around me got it" |
| No symptoms | 26 (14.3) | "No, because I didn't notice any symptoms amongst myself or my family." |
| No need | 26 (14.3) | "I have not because I have been incredibly cautious and have not felt the need to" |
| **5. Would access to free COVID-19 self-tests be helpful to you? Why or why not? (n = 932)** | | |
| Yes | 629 (67.5) | |
| Maybe | 56 (6.0) | "If it's effective" |
| No | 199 (21.4) | |
| *Reasons helpful*** | | |
| Easier | 118 (17.2) | "Yes, it would be more convenient" |
| Peace of mind | 88 (12.8) | "yes, i would feel more comfortable going around" |
| Proactive | 87 (12.7) | "yes! i would test myself before i went out if it were that easy" |
| Cost | 80 (11.7) | "Yes because I don't have enough money to pay for them" |
| Accessibility | 58 (8.5) | "Yes hard to find a site same day" |
| *Reasons not helpful*** | | |
| Other options | 45 (17.6) | "Not really, I have a few local places I could go to get tested" |
| Wouldn't use | 35 (13.7) | "Not really I wouldn't use them in reality" |
| Not accurate | 27 (10.6) | "No, they do not work" |
| No need | 26 (10.2) | "No because I do not have to get tested and have not for months" |
| Vaccinated | 18 (7.1) | "no, i'm fully vaccinated" |

*Percentages may not add to 100% as codes are not mutually exclusive, not all codes are presented, and percentages may represent a subgroup

** Percentages for reasons for testing (Q2 and Q3) were calculated out of responses that indicated yes or depends; percentages for reasons against testing (Q2 and Q3) were calculated out of responses that indicated no or depends.

*** Percentages for reasons helpful (Q5) were calculated out responses that indicated yes or maybe; percentages for reasons not helpful (Q5) were calculated out of responses that indicated no or maybe.

asymptomatic if they do not display fever or signs if respiratory illness although asymptomatic individuals can still contribute [28] to the spread of the virus. Although the largest spike in cases [29] came after the introduction of vaccines [30], many institutional testing measures were mollified at this time as well, reducing the availability of an important avenue for screening. In 2020, it was estimated that, 63% [31] of those who completed high school attended college, and estimated that 12.2% [32] of those in our study age group is part of the workforce, indicating a high potential for transmission of COVID both on college campuses and in the workforce. A qualitative survey with this sample can give policymakers unique insights into the facilitators and barriers to ongoing testing as a means to slow COVID-19 transmission. As we see a reduction in testing mandates, it will be important to ensure our schools and workplaces are equipped with other modalities to assist in mitigating the spread. Analyzing how to successfully implement strategies such as pooled testing—testing an entire classroom together to increase odds of detecting infection [33], similarly testing wastewater and air filter en masse for the shed virus, test-to-stay and test-to-play—requiring a negative test 72 hours prior to a school sporting event [34], and vigilant contact tracing demonstrated in Australian schools [35] can be critical in the strive to return to in person settings [36]. If rates increase or another pandemic occurs, school and workplace requirements may be effective in influencing youth behaviors, though the burden these requirements may impose on youth should be carefully considered.

As we see a nationwide decrease in institutional COVID-19 testing and an end to the public health emergency, at-home tests may likely be a solution we rely on to increase and maintain

access to testing. Most respondents reported home testing would be valuable with reasons including that it would be easier, save on time, and would bring a sense of peace going into gatherings. Current testing options include molecular (PCR) and antigen (rapid) tests [1]. Both molecular and antigen tests are often covered at no cost by insurance providers. Free government sponsored orders [3, 37] via USPS for at-home rapid tests for households have also been reissued three times during the rise in COVID-19 cases. Given this response, it is evident there is a desire and clear need for self tests. As youth still may be limited in their ability to access these resources, the "MI Backpack Home Test Program" [38] exemplifies how this might be achieved. Sponsored by the Michigan Department for Health and Human Services (MDHHS), the program provides a kit to registered students which includes two at-home antigen tests and additional educational materials. In January 2022, the Biden administration committed to sending millions of free tests to schools monthly [39]. Granting access to home COVID-19 tests through methods such as these may allow individuals, especially those without access to transportation for testing with a healthcare worker, to quickly respond to suspected infections and make decisions that can reduce the incidence of COVID-19.

The study was limited by several factors. The study operated via text message response which does not allow for clarification of responses. Additionally, while the study did include a diverse population, it over-represents non-Hispanic white and underrepresents female, Hispanic, and non-Hispanic black populations relative to current nationwide demographics which limits its generalizability [20]. Also, while responses from youth described their testing behaviors, we did not inquire into their corresponding behaviors based on the results of those tests. For example, would a positive test influence a decision to attend a social gathering? In addition, the study was conducted at a time when at home testing was still a novel concept, meaning that participants may not have yet had sufficient knowledge or context to provide their thoughts and opinions on at-home testing. Guided by future studies, such questions and may more directly measure the impact of testing.

The majority of youth in our nationwide sample reported that they would likely test for COVID-19 especially if they felt they were exposed, though less likely if they are vaccinated. However, youths' approach to testing largely does not adhere to current CDC guidelines on who and when to test [1]. While this does mirror attitudes toward testing at large [40–42], youth have limited access to home testing for COVID-19, so are reasonably interested in increased access. Youth-centered communications regarding testing recommendations and increasing availability of tests, specifically for youth, may reduce the spread of COVID-19 among youth and inform future pandemic recommendations.

## Acknowledgments

The authors would like to thank the MyVoice participants for their continued participation and candor in sharing their experiences and perspectives as well as Katie Grode for her assistance in submitting this manuscript.

## Author Contributions

**Conceptualization:** Tammy Chang.

**Data curation:** Caleb Arthur, Kathleen Abenes, Marika Waselewski, Samantha A. Chuisano.

**Formal analysis:** Caleb Arthur, Kathleen Abenes, Marika Waselewski, Samantha A. Chuisano.

**Investigation:** Tammy Chang.

**Methodology:** Marika Waselewski, Tammy Chang.

**Project administration:** Tammy Chang.

**Supervision:** Tammy Chang.

**Writing – original draft:** Caleb Arthur.

**Writing – review & editing:** Caleb Arthur, Kathleen Abenes, Marika Waselewski, Samantha A. Chuisano, Tammy Chang.

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
