## [Decision Letter · Decision Letter 0]

17 Apr 2023

PONE-D-23-02617Youths’ Perceptions and Behaviors on COVID-19 TestingPLOS ONE

Dear Dr. Arthur,

Thank you for submitting your manuscript to PLOS ONE. After careful consideration, we feel that it has merit but does not fully meet PLOS ONE’s publication criteria as it currently stands. Therefore, we invite you to submit a revised version of the manuscript that addresses the points raised during the review process.

This is important research that would add to the knowledge base of COVID-19 and inform future responses to health emeregncies. However, authors need to consider substantial revisions to the manuscript to address both reviewers' comments about methodology and discussion.We would expect a more thorough proofreading to improve the writing to a publishable standard. 

We look forward to receiving your revised manuscript.

Kind regards,

Phyllis Lau, PhD

Academic Editor

PLOS ONE

Journal Requirements:

Additional Editor Comments:

This is important research that would add to the knowledge base of COVID-19 and inform future responses to health emeregncies. However, authors need to consider substantial revisions to the manuscript to address both reviewers' comments about methodology and discussion.We would expect a more thorough proofreading to improve the writing to a publishable standard.

Reviewers' comments:

Reviewer's Responses to Questions

**Comments to the Author**

1. Is the manuscript technically sound, and do the data support the conclusions?

Reviewer #1: Yes

Reviewer #2: Yes

2. Has the statistical analysis been performed appropriately and rigorously? 

Reviewer #1: No

Reviewer #2: No

3. Have the authors made all data underlying the findings in their manuscript fully available?

Reviewer #1: Yes

Reviewer #2: Yes

4. Is the manuscript presented in an intelligible fashion and written in standard English?

Reviewer #1: Yes

Reviewer #2: Yes

5. Review Comments to the Author

Reviewer #1: I enjoyed reading this well written article that addresses the problem of containing the spread of Covid in young people. Thank you for the opportunity to provide a review of your paper.

The abstract was clear and represented the article well, but the methods section will need revision considering my comments below.

The introduction was well written and justified the research well, placing it in context of existing literature.

The methods section needs revision.

The sample was described well.

The descriptive analysis of the sample should be accompanied by a comparison with US demographic data for your sample’s age group to demonstrate the presence or otherwise of ascertainment bias. In the limitations section of the discussion, you mentioned that the sample was not representative. How was the sample different, and how does this influence the interpretation of your findings?

The analytic approach to the comments in the survey was described as ‘thematic analysis’ in the abstract and the reference for the coding approach within the methods cites an article that describes qualitative thematic analysis. Your approach was more in keeping with ‘content analysis’, where data are coded according to carefully developed criteria allowing descriptive analysis of the coded data. In classical qualitative analysis using thematic analysis, as described in the article you cited, themes are identified within texts to answer a research question, but the findings cannot be quantified. There are also mandatory requirements for ensuring the trustworthiness of qualitative research.

Here is an article that better aligns with the approach you took:

Hsieh HF, Shannon SE. Three approaches to qualitative content analysis. Qualitative health research. 2005 Nov;15(9):1277-88.

Or this book

Neuendorf, K., 2017. The Content Analysis Guidebook. Second Thousand Oaks, CA: SAGE Publications, Inc.

You need to rewrite your methods to align with the methodology you used. Content analysis also allows you to look for associations between your codes and your demographic data which might strengthen your recommendations. For example, was there an association between response types and gender, region, or education level?

The discussion was well written and answered your research question. The implications of your research were valid and well-argued, but perhaps you could add more if you knew how to target the interventions.

This paper, with revision to the description of the methods, is worthy of publication as it provides a valuable insight into the perceptions of young people about Covid testing.

Reviewer #2: Thank you for the opportunity to review this paper. It is an interesting study that will likely add to literature around the behavioural aspects of pandemic responsiveness in the younger age groups.

While I think it is important to publish these findings I think that there is some work to do first

Major Comments:

1. The language and grammar is generally good and it is succinctly and clearly written, without over complicating things. However, it can sound a little parochial in places (e.g. 'gotten'), phrasing is sometimes a little odd and would overall benefit from some experienced peer review at the authors institute to tighten this up.

2. The data is not consistently analysed. The table headers suggests that the data presented is taken from the total n with percentages calculated using the total response for each question as a denominator. However, this appears to change with each category/question. While I think the sub categories (reasons) for the yes/no responses should be calculated using the total of that yes/no response (rather than the whole question response n value), this is not consistently applied and is not apparent from the text. In some cases the NO and MAYBE responses are added together to give the denominator and other times this is not the case.

You must address this either through re-analysing with consistent methodology or make this clear to the reader so the the data are contextualised. By using the yes/no helpful/not helpful response n value you will provide a clearer idea of the break down. You may also like to look at ways of illustrating this data graphically rather than within a table to make it clearer and more engaging for readers.

3. Discussion. Many assumptions are made in the discussion, without any evidence or supporting data. It feels rushed. For example on line 157-159 the statement "COVID-19 cases can occur in those who have been vaccinated, and education efforts among youth that explicitly highlight testing recommendations among both vaccinated and unvaccinated individuals could impact youth testing behavior" requires supporting evidence.

Home testing. Line 180 onwards. Literature is available on home testing and many countries have successfully implemented this in schools (e.g Australia) , I would recommend that the authors include data evidence and literature surrounding this.

Please state which guidelines are being referred to on line 205.

there needs to be greater acknowledgement previous work, nationally and internationally in this field of COVID19 research.

Minor Issues:

1. Phrasing on Lines 61 and 177. Please review and tidy up the language.

2. Line 71. " Many youth are often asymptomatic when positive for COVID " - what is the figure?

3. Though it states in methods that MyVoice is described elsewhere a summary of the app would be helpful for basic context. e.g. age range, SES, length of engagement with app etc. were the respoinses representative of most users?

4. Table 1. Why % or Mean SD in header?

5. Line 117 please amend to "symptoms consistent with or suspicious for, COVID-19 "

6. Line 127-129. Please amend to include 'from' - %). There were more responses of from those who would not get tested before a social gathering (30.5%, 292 of 958) than from those who would ..."

7. A per major comment 2. Line 130 - I would be inclined quote the figures as a % of those that said NO to testing before social gatherings, rather than the total responses as this only applies to this group

8. Line 145. please amend to replace 'though' with 'and': "also would get tested after a gathering, and fewer reported that they would..."

9. Line 198-201 - Please give examples of what you mean here, for greater context.

Many thanks

6. PLOS authors have the option to publish the peer review history of their article (what does this mean?). If published, this will include your full peer review and any attached files.

Reviewer #1: No

Reviewer #2: No

---

## [Author Response · Author response to Decision Letter 0]

21 Jun 2023

EDITOR’S SPECIFIC COMMENTS:

Comment 1: Please ensure that your manuscript meets PLOS ONE’s style requirements, including those for file naming. The PLOS ONE style templates can be found at

Response 1: In accordance with the journal’s style requirements, headings have been changed to font size 18. Data is available upon request after completion of a data sharing agreement. Please contact irbmed@umich.edu or 734-763-4768 / (Fax 734-763-1234); 2800 Plymouth Road, Building 520, Room 3214, Ann Arbor, MI 48109-2800 to request access to data. Title, author, and affiliations have been formatted to conform to PLOS ONE guidelines.

Comment 2. You indicated that you had ethical approval for your study. In your Methods section, please ensure you have also stated whether you obtained consent from parents or guardians of the minors included in the study or whether the research ethics committee or IRB specifically waived the need for their consent.

Response 2: Line 85-87 have been added: “Written informed consent was obtained electronically from all participants; given minimal risk, parental consent was waived by the IRB.”

Comment 3: Please provide additional details regarding participant consent. In the ethics statement in the Methods and online submission information, please ensure that you have specified (1) whether consent was informed and (2) what type you obtained (for instance, written or verbal, and if verbal, how it was documented and witnessed). If your study included minors, state whether you obtained consent from parents or guardians. If the need for consent was waived by the ethics committee, please include this information.

Response 3: As above, lines 85-87 have been added to specify that written informed consent was obtained online for all participants with parent consent waived by IRB. Line 95-98 now reflects that the data has been anonymized: “Using an inductive approach, two team members individually reviewed fully anonymized youth responses for each question and identified common responses that were then used to develop a codebook.”

Comment 4: We note that you have indicated that data from this study are available upon request. PLOS only allows data to be available upon request if there are legal or ethical restrictions on sharing data publicly. For more information on unacceptable data access restrictions, please see http://journals.plos.org/plosone/s/data-availability#loc-unacceptable-data-access-restrictions.

Response 4: Our study data is protected by an NIH Certificate of Confidentiality which requires a signed data sharing agreement to be completed before data can be accessed. Please contact Dr. Tammy Chang to request access to the data. This has been added to the cover letter.

Comment 5: PLOS requires an ORCID iD for the corresponding author in Editorial Manager on papers submitted after December 6th, 2016. Please ensure that you have an ORCID iD and that it is validated in Editorial Manager. To do this, go to ‘Update my Information’ (in the upper left-hand corner of the main menu), and click on the Fetch/Validate link next to the ORCID field. This will take you to the ORCID site and allow you to create a new iD or authenticate a pre-existing iD in Editorial Manager. Please see the following video for instructions on linking an ORCID iD to your Editorial Manager account: https://www.youtube.com/watch?v=_xcclfuvtxQ

Response 5: ORCID IDs for corresponding author has been added.

Comment 6: This is important research that would add to the knowledge base of COVID-19 and inform future responses to health emergencies. However, authors need to consider substantial revisions to the manuscript to address both reviewers’ comments about methodology and discussion. We would expect a more thorough proofreading to improve the writing to a publishable standard.

Response 6: The methodology has been updated and discussion thoroughly reviewed and edited as above. The manuscript has been reviewed in total for clarity.

REVIEWER’S SPECIFIC COMMENTS:

Reviewer #1

Comment 6: The descriptive analysis of the sample should be accompanied by a comparison with US demographic data for your sample’s age group to demonstrate the presence or otherwise of ascertainment bias.

Response 7: Lines 111-113 have been added to reflect US demographic: “For comparison, 49.5% of the nation identifies as male, 59.3% as non-Hispanic white, and 63.5% as completing some college or technical school. The Census Bureau does not provide data for those in suburban areas, but it is estimated 80.0% live in urban settings while 20.0% are within rural settings.”

Comment 8: In the limitations section of the discussion, you mentioned that the sample was not representative. How was the sample different, and how does this influence the interpretation of your findings?

Response 8: Lines 219-222 have been modified to reflect the differences in representation of our sample from US demographics: “Additionally, while the study did include a diverse population, it is not reflective of current nationwide demographics“ changed to “Additionally, while the study did include a diverse population, it over-represents non-Hispanic white and underrepresents female, Hispanic, and non-Hispanic black populations relative to current nationwide demographics which limits its generalizability.”

Comment 9: The analytic approach to the comments in the survey was described as ‘thematic analysis’ in the abstract and the reference for the coding approach within the methods cites an article that describes qualitative thematic analysis. Your approach was more in keeping with ‘content analysis’, where data are coded according to carefully developed criteria allowing descriptive analysis of the coded data. In classical qualitative analysis using thematic analysis, as described in the article you cited, themes are identified within texts to answer a research question, but the findings cannot be quantified. There are also mandatory requirements for ensuring the trustworthiness of qualitative research.

Here is an article that better aligns with the approach you took:

Hsieh HF, Shannon SE. Three approaches to qualitative content analysis. Qualitative health research. 2005 Nov;15(9):1277-88.

Or this book

Neuendorf, K., 2017. The Content Analysis Guidebook. Second Thousand Oaks, CA: SAGE Publications, Inc.

You need to rewrite your methods to align with the methodology you used. Content analysis also allows you to look for associations between your codes and your demographic data which might strengthen your recommendations. For example, was there an association between response types and gender, region, or education level?

Response 9: Thank you for these wonderful references and we agree. We have updated our methods to state “content analysis” and included the references you provided. 

Comment 10: The discussion was well written and answered your research question. The implications of your research were valid and well-argued, but perhaps you could add more if you knew how to target the interventions.

Response 10: Target intervention including pooled testing, testing-to-play, wastewater and air filter testing have been added to lines 191-196: “Analyzing how to successfully implement strategies such as pooled testing—testing an entire classroom together to increase odds of detecting infection, similarly testing wastewater and air filter en masse for the shed virus, test-to-stay and test-to-play—requiring a negative test 72 hours prior to a school sporting event, and vigilant contact tracing demonstrated in Australian schools can be critical in the strive to return to in person settings”. Target intervention of increased test production has been added to lines 212-213: “In 2022, the Biden administration committed to sending millions of free tests to schools monthly.”

Reviewer #2: 

Comment 11: The language and grammar is generally good and it is succinctly and clearly written, without over complicating things. However, it can sound a little parochial in places (e.g. ‘gotten’), phrasing is sometimes a little odd and would overall benefit from some experienced peer review at the authors institute to tighten this up.

Response 11: The manuscript has been reviewed in total and edited to improve readability.

Comment 12: The data is not consistently analysed. The table headers suggests that the data presented is taken from the total n with percentages calculated using the total response for each question as a denominator. However, this appears to change with each category/question. While I think the sub categories (reasons) for the yes/no responses should be calculated using the total of that yes/no response (rather than the whole question response n value), this is not consistently applied and is not apparent from the text. In some cases the NO and MAYBE responses are added together to give the denominator and other times this is not the case.

You must address this either through re-analysing with consistent methodology or make this clear to the reader so the data are contextualised. By using the yes/no helpful/not helpful response n value you will provide a clearer idea of the break down. You may also like to look at ways of illustrating this data graphically rather than within a table to make it clearer and more engaging for readers.

Response 12: Subcategories for all data within the text has been added. Question, “Would access to free COVID-19 self-tests be helpful to you?” has now been calculated with a consistent denominator. Lines 123-156 have been modified to consistently reflect the subgroups form which the statistics have been calculated. Table 2 footer has been updated to clarify calculations: 

“**Percentages for reasons for testing (Q2 and Q3) were calculated out of responses that indicated yes or depends; percentages for reasons against testing (Q2 and Q3) were calculated out of responses that indicated no or depends. 

*** Percentages for reasons helpful (Q5) were calculated out responses that indicated yes or maybe; percentages for reasons not helpful (Q5) were calculated out of responses that indicated no or maybe." 

We have elected to present the data in a table to more efficiently present the quotes, though will defer to editors if they suggest alternative options. 

Comment 13: Discussion. Many assumptions are made in the discussion, without any evidence or supporting data. It feels rushed. For example on line 157-159 the statement “COVID-19 cases can occur in those who have been vaccinated, and education efforts among youth that explicitly highlight testing recommendations among both vaccinated and unvaccinated individuals could impact youth testing behavior” requires supporting evidence.

Response 13: The line, “COVID-19 cases can occur in those who have been vaccinated, and education efforts among youth that explicitly highlight testing recommendations among both vaccinated and unvaccinated individuals could impact youth testing behaviors” were deleted given redundancy to lines 173-175.

Lines 173-175 were amended for clarity, originally reading, “Education on appropriate testing at the moment youth are getting vaccinated may represent a teachable moment for parents and youth alike. Finally, communication that is targeted towards youth themselves may promote more testing as greater access to home test kits are available” changed to “Education on appropriate timing of testing at the moment youth are getting vaccinated may represent a teachable moment for parents and youth alike. Finally, communication that is targeted towards youth themselves may promote more testing as greater access to self-test kits are available” with references added. 

Comment 14: Home testing. Line 180 onwards. Literature is available on home testing and many countries have successfully implemented this in schools (e.g Australia) , I would recommend that the authors include data evidence and literature surrounding this.

Response 14: Successful strategies of including pooled testing, testing-to-play, wastewater and air filter testing have been added to lines 191-196: “Analyzing strategies how strategies such as pooled testing—testing an entire classroom together to increase odds of detecting infection, similarly testing wastewater and air filter en masse for the shed virus, test-to-stay and test-to-play—requiring a negative test 72 hours prior to a school sporting event, and vigilant contact tracing demonstrated in Australian schools can be implemented successfully can be critical in the strive to return to in person settings”.

Comment 15: Please state which guidelines are being referred to on line 205.

Response 15: Lines 253-254, “However, youths’ use of tests largely does not adhere to current guidelines” changed to “However, youths’ approach to testing largely does not adhere to current CDC guidelines on who and when to test.”

Comment 16: there needs to be greater acknowledgement previous work, nationally and internationally in this field of COVID19 research.

Response 16: Several citations have been added, notably in lines 233-234, originally, “Youth are interested in increased access to home testing,” amended to, “While this does mirror attitudes toward testing at large, youth are limited in access so are reasonably interested in increased access to home testing, so are reasonably interested in increased access.” 

Comment 17: Phrasing on Lines 61 and 177. Please review and tidy up the language.

Response 17: Line 56-57, “In 2022 guidelines were changed to reflect a community’s transmission level, advising testing based on hospitalization rates of COVID” has been amended for clarity: “Later that year, the CDC changed its guidelines, which advised testing based on hospitalization rates of COVID instead of community levels.”

Lines 196-198, “If rates increase or another pandemic occurs, school and workplace requirements may be effective in influencing youth behaviors, though careful consideration of the burden and experiences of youth of these requirements should be taken,” has been amended for clarity: “If rates increase or another pandemic occurs, school and workplace requirements may be effective in influencing youth behaviors, though the burden these requirements may impose on youth should be carefully considered.”

Comment 18: Line 71. " Many youth are often asymptomatic when positive for COVID " - what is the figure?

Response 18: Lines 67-69 has been amended to demonstrate this figure: “Many youth are often asymptomatic when positive for COVID, and their behaviors influence the transmission of COVID in their communities,” changed to “Many youth are often asymptomatic when they contract COVID demonstrated by a 2022 meta-analysis showing 43.75% of those testing positive under 20 years of age were asymptomatic.”

Comment 19: Though it states in methods that MyVoice is described elsewhere a summary of the app would be helpful for basic context. e.g. age range, SES, length of engagement with app etc. were the responses representative of most users?

Response 19: MyVoice has now been described in lines 77-82. “MyVoice recruitment occurs online via targeted social media advertisements based on weighted benchmarks from the American Community Survey. Ads are targeted based on age, gender, race and ethnicity, and region of residence to encourage a diverse cohort. Cohorts range typically between 1000 to 1200 participants. Individuals ages 14-24 years with literacy in the English language and access to a phone with texting capabilities are encouraged to apply.”

Comment 20: Table 1. Why % or Mean SD in header?

Response 20: Mean SD are in the header as they describe how age is reported.

Comment 21: Line 117 please amend to "symptoms consistent with or suspicious for, COVID-19 "

Response 21: This line, now 125-126, has been amended.

Comment 22: Line 127-129. Please amend to include 'from' - %). There were more responses of from those who would not get tested before a social gathering (30.5%, 292 of 958) than from those who would ..."

Response 22: This line, now 141-144, “There were more responses of those who would not get tested before a social gathering (30.5%, 292 of 958) than those who would not get tested after an event (15.7%, 146 of 931) with vaccination status the most cited as the reason (9.0% of 958 and 5.6% of 931, respectively),” has been amended to “There were more responses from those who would not get tested before a social gathering (30.5%, 292 of 958) than those who would not get tested after a social event (15.7%, 146 of 931) with vaccination status the most commonly cited reason (18.5%, 86 of 465 and 19.6%, 52 of 265, respectively).”

Comment 23: A per major comment 2. Line 130 - I would be inclined quote the figures as a % of those that said NO to testing before social gatherings, rather than the total responses as this only applies to this group

Response 23: Lines 140-149 now reflect percentages of subgroups rather than of total responses: “A greater number of participants endorsed they would get tested after a social gathering (70.6%) than the total number for before an event (50.9%, 488 of 958). There were more responses of those who would not get tested before a social gathering (30.5%, 292 of 958) than those who would not get tested after an event (15.7%, 146 of 931) with vaccination status most cited as the reason (9.0% of 958 and 5.6% of 931, respectively). Additional reasons for not testing before included lack of involvement in social gatherings (2.7%, 26 of 958), “feeling fine” (4.8%, 46 of 958), and thinking the test is unnecessary (6.7%, 31 of 465). Other reasons for not testing after included mask use in public spaces (5.2%, 11 of 931) and thoughts that testing was pointless or assurance they would know when they were sick (2.0%, 19 of 931),” 

changed to: “A greater number of participants endorsed they would get tested after a social gathering (70.6%, 657 of 931) than the total number for before an event (50.9%, 488 of 958). There were more responses from those who would not get tested before a social gathering (30.5%, 292 of 958) than those who would not get tested after an event (15.7%, 146 of 931) with vaccination status most cited as the reason (18.5%, 86 of 465 and 19.6%, 52 of 265, respectively). Additional reasons for not testing before included lack of involvement in social gatherings (5.6%, 26 of 465), “feeling fine” (9.9%, 46 of 465), and thinking the test is unnecessary (6.7%, 31 of 465). Other reasons for not testing after included mask use in public spaces (14.2%, 11 of 265) and thoughts that testing was pointless or assurance they would know when they were sick (7.2%, 19 of 265).”

Comment 24: Line 145. please amend to replace 'though' with 'and': "also would get tested after a gathering, and fewer reported that they would..."

Response 24: This line, now 161, has been amended.

Comment 25: Line 198-201 - Please give examples of what you mean here, for greater context.

Response 25: Added lines 275-276 for context: “For example, would a positive test influence a decision to attend a social gathering? In addition, the study was conducted at a time when at home testing was still a novel concept, meaning that participants may not have yet had sufficient knowledge or context to provide their thoughts and opinions on at-home testing. Guided by future studies, such questions and may more directly measure the impact of testing.

---

## [Decision Letter · Decision Letter 1]

1 Aug 2023

Youths’ Perceptions and Behaviors on COVID-19 Testing

PONE-D-23-02617R1

Dear Dr. Arthur,

We’re pleased to inform you that your manuscript has been judged scientifically suitable for publication and will be formally accepted for publication once it meets all outstanding technical requirements.

Kind regards,

Phyllis Lau, PhD

Academic Editor

PLOS ONE

Additional Editor Comments (optional):

Reviewers' comments:

Reviewer's Responses to Questions

**Comments to the Author**

1. If the authors have adequately addressed your comments raised in a previous round of review and you feel that this manuscript is now acceptable for publication, you may indicate that here to bypass the “Comments to the Author” section, enter your conflict of interest statement in the “Confidential to Editor” section, and submit your "Accept" recommendation.

Reviewer #1: (No Response)

2. Is the manuscript technically sound, and do the data support the conclusions?

Reviewer #1: Yes

3. Has the statistical analysis been performed appropriately and rigorously? 

Reviewer #1: Yes

4. Have the authors made all data underlying the findings in their manuscript fully available?

Reviewer #1: Yes

5. Is the manuscript presented in an intelligible fashion and written in standard English?

Reviewer #1: Yes

6. Review Comments to the Author

Reviewer #1: Thank you for your responses to my suggestions. I did note a typo on p2 L46 the word Covid-19 was missing.

Otherwise the paper is worthy of publication and will be of interest to public health and adolescent physicians for managing this and future pandemics.

7. PLOS authors have the option to publish the peer review history of their article (what does this mean?). If published, this will include your full peer review and any attached files.

Reviewer #1: **Yes: **Jenny McDonald

---

## [Editor Report · Acceptance letter]

3 Aug 2023

PONE-D-23-02617R1 

Youths’ perceptions and behaviors on COVID-19 testing 

Dear Dr. Arthur:

I'm pleased to inform you that your manuscript has been deemed suitable for publication in PLOS ONE. Congratulations! Your manuscript is now with our production department. 

Kind regards, 

on behalf of

Dr. Phyllis Lau 

Academic Editor

PLOS ONE